# Using Citizen Science to Explore Neighbourhood Influences on Ageing Well: Pilot Project

**DOI:** 10.3390/healthcare7040126

**Published:** 2019-11-01

**Authors:** Helen Barrie, Veronica Soebarto, Jarrod Lange, Fidelma Mc Corry-Breen, Lauren Walker

**Affiliations:** 1The Hugo Centre for Migration and Population Research, School of Social Sciences, University of Adelaide, Adelaide, South Australia 5005, Australia; jarrod.lange@adelaide.edu.au (J.L.); fidelma.breen@adelaide.edu.au (F.M.C.-B.); lauren.walker@adelaide.edu.au (L.W.); 2School of Architecture and Built Environment, University of Adelaide, Adelaide, South Australia 5005, Australia; veronica.soebarto@adelaide.edu.au

**Keywords:** citizen science, built environment, older people, urban neighbourhoods, GIS, spatial, Australia

## Abstract

Outdoor and indoor environments impact older people’s mobility, independence, quality of life, and ability to “age in place”. Considerable evidence suggests that not only the amount, but also the quality, of public green spaces in the living environment is important. The quality of public green spaces is mostly measured through expert assessments by planners, designers and developers. A disadvantage of this expert-determined approach is that it often does not consider the appraisals or perceptions of residents. Daily experience, often over long periods of time, means older residents have acquired insider knowledge of their neighbourhood, and thus, may be more qualified to assess these spaces, including measuring what makes a valued or quality public green space. The aim of this Australian pilot study on public green spaces for ageing well was to test an innovative citizen science approach to data collection using smart phones. “Senior” citizen scientists trialed the smart phone audit tool over a three-month period, recording and auditing public green spaces in their neighbourhoods. Data collected included geocoded location data, photographs, and qualitative comments along with survey data. While citizen science research is already well established in the natural sciences, it remains underutilised in the social sciences. This paper focuses on the use of citizen science with older participants highlighting the potential for this methodology in the fields of environmental gerontology, urban planning and landscape architecture.

## 1. Introduction

The design and delivery of quality public green spaces that promote health and wellbeing, social engagement with others and engagement with the environment is a key challenge in our rapidly growing, and increasingly population-dense cities. As cities become denser, incorporating quality public green spaces becomes more important than ever [1,2,3,4]. A greater understanding of how these spaces should be designed is needed to support the physical, mental and social health of individuals. By 2030 two-thirds of the world’s population will be living in cities and, in many of these cities, at least a quarter of the population will be aged 65 plus years [5]. Cities, particularly our inner-city built environments, are spaces that are usually imagined, planned and structured with a younger, working age demographic in mind. This project was conducted in South Australia—chiefly in Greater Metropolitan Adelaide. South Australia (SA) is the oldest of the mainland states of Australia, with 37.8% of its population aged 50 + years [6]. Thus, planning our cities and neighbourhoods for an older population is an extant reality; yet older people are not typically incorporated into the mainstream of thinking and planning around urban development, public open and green space environments.

For the purposes of this project, public green spaces included any public or civic space that included forms of vegetation (grass, trees, gardens, formal planting and/or natural bushland) that are maintained by local, state or federal governments or private organisations but are accessible to all members of the public. Public green spaces include parks, gardens, reserves, sporting fields, walking trails, riparian areas such as riverbanks, trees and verges as part of streetscapes, and courtyards or ‘green walls’ that form parts of public buildings. These public green spaces vary in size, presentation, quality and purpose and have a diversity of vegetation cover and species.

This pilot study aimed to test a new smart phone-based audit tool using an innovative methodology—citizen science—in order to explore how and why older people engage with public green spaces. The pilot project presented in this paper was built upon the premise that it is important to understand the relationship between older people and these public green spaces beyond simple utilisation of the physical space. While the fabric of the physical space (such as the housing environment, surrounding neighbourhood, public buildings as well as the public green spaces) is important, we also need to consider the interplay of these built environment elements with health, wellbeing, social connectedness and civic engagement, as well as mediums for maintaining autonomy and independence. We need to imagine how the built environment, including public green spaces, can become enablers of ageing well, and this needs to be examined from the insider perspective of the older person.

## 2. Background

Standards and guidelines have been developed to address principles to improve public spaces, neighbourhoods, buildings and constructions to ensure that older people can fully utilise those spaces. The World Health Organization (WHO) has developed a guideline for achieving “age-friendly cities”, or cities that encourage “active ageing by optimizing opportunities for health, participation and security in order to enhance quality of life as people age” [7]. The guideline focuses on eight main topic areas that must be addressed: public spaces and buildings, transportation, housing, social participation, respect and social inclusion, civic participation and employment, communication and information, and community support and health services [7]. Outdoor and indoor environments are considered to have major impacts on older people’s mobility, independence and quality of life in cities, and particularly on their ability to “age in place” (or to live as long as possible in one’s own home). The guideline specifies 11 areas to be addressed in the topic of “outdoor spaces and buildings” including pleasant and clean environments, the importance of green spaces, somewhere to rest, age-friendly pavements, safe pedestrian crossings, accessibility, secure environment, barrier-free buildings, and adequate public toilets [7].

Local governments around the world have developed their own policies, plans, programs and services to improve the “age-friendliness” of their cities by adopting the WHO guideline [7]. Researchers have previously conducted studies to audit neighbourhoods [8,9], and developed tools to conduct the audits [10]. It is, however, questionable whether this expert-determined approach reflects the appraisals or perceptions of the older residents about their own environment. As older people have a tendency to live in the same place, often over long periods of time, they are likely to have first-hand or insider knowledge about their neighbourhood, and thus, may be more qualified to assess these spaces and understand what makes a valued or quality public green space [11,12]. Citizen science, an emerging methodological approach in the social sciences, offers insightful opportunities for creating strong appraisals of age friendly adaptations of the built environment. This enables developers, planners and academics to better understand what really makes a community or neighbourhood age friendly from the perspective of the older person who uses that space.

In recent years, we have witnessed important shifts in the relationship between science and society. The discussion has moved away from a classical “public understanding of science” approach, aiming at transferring knowledge about scientific processes to the public, to a “science in society” approach where the public is engaged in the production of science. One practical approach to engaging citizens in the scientific process is co-design, and another is “citizen science” [13].

The term citizen science is used in different ways. For the purpose of this research, we view citizen science as a partnership between professional researchers and volunteers in which the volunteers implement tasks which have traditionally been implemented by scientists [14,15]. This cooperation is meant to serve two goals. First, it should create new scientific insights, most importantly by gathering large-scale or hidden data, which the researchers alone could not access or generate. Second, the partnership should produce an educational outcome for the participants, such as increasing knowledge and scientific interest.

Citizen science employs a cooperative approach to research. There are three possible models of cooperation that have been identified: (1) contributive, (2) collaborative, and (3) co-created [15,16]. In the contributive model, volunteers (the citizen scientists) contribute to data collection only. This may also be called “crowd sourcing” data. Note that this is different from researchers merely collecting data from or about participants. In the collaborative model, the citizen scientists may also be engaged in data analysis and interpretation. In the co-created model, the citizen scientists are involved in all stages of the scientific process, including assisting in defining the research questions and developing the research design [14,15,16].

In this project, instead of basing the evaluation of public green spaces on the researcher’s value judgement, we trialed the citizen science approach with a co-created model. As citizen scientists, older people not only collected data but were also engaged in preliminary analysis of the data and, most importantly, contributed feedback and ideas on the methods, process, audit tool and the design of the proposed larger project.

## 3. Materials and Methods

### 3.1. Development of Audit Tools

As well as trialing the citizen science approach to evaluate public green spaces, the pilot project developed and tested audit tools to be used by the citizen scientists to evaluate these spaces around their own neighbourhoods.

Two main audit tools were developed: (1) an online tool for a smart phone and (2) a field note booklet used in conjunction with a disposable digital camera that replicates the smart phone audit tool. The audit tools were developed through a combination of previous work by the research team in the field of built environment (yet to be published) and review of current literature. The tools developed for the pilot have not been tested for reliability or validity at this stage as this work is ongoing. Development of the audit tools included a set of printed instructions—including some safety tips regarding using a mobile phone while walking, and privacy legislation regarding taking photographs. The audit tools were trialed by the research team during development before participants were recruited to take part in the pilot study. Participants selected either the mobile-based or paper-based tool depending on their preference and comfort with, and access to, the appropriate technology.

The online tool was hosted on the ESRI platform, Survey123™ [17]. The audit tool allowed participants to record their experiences and perceptions in using public spaces in the course of their normal day-to-day activities, chiefly by responding to Likert scale questions relating to several key areas. They are: (1) the state of their general health and well-being at the time of the audit; (2) the space itself, including overall visual perception, state of cleanliness, feeling of safety/security, user friendliness, comfort, noise and busyness, lighting quality, and greenness; (3) the nature of the visit to the space, including purpose of visit (e.g., to relax, to meet with people, to exercise), average frequency of visits, length of time of this visit, mode of travel (e.g., walking, driving or by public transport); and (4) facilities available in that location (e.g., public toilets, seating, shade, drinking water availability). Critically for data analysis, the ESRI Survey123™ audit tool includes a location finder question, allowing each audit to be geocoded. The tool also allowed for uploading two photographs and a 250-character open text box for additional comments about the audit as additional qualitative data (see Figure 1). Each time a citizen scientist submitted the audit survey, the data were sent to the Hugo Centre’s ESRI online cloud-based service. This enabled the research team to validate the quality and frequency of the audits completed in real-time and offer individual support to participants regarding use of the auditing tool.

### 3.2. Recruitment and Training

Older citizen scientists were recruited through advertisements in the “Weekend Plus” magazine, an online weekly magazine produced by the Office for Ageing Well (Previously the Office for the Ageing) available to South Australians eligible for a Seniors Card. Recruitment also took place through the newsletter of the Adelaide City Council (ACC) and on the “Plug-In” community website of the Council of the Ageing (COTA) SA. The latter was the most successful recruitment route with most senior citizen scientists becoming part of the research team through the COTA site. Human research ethics approval was obtained from the Human Research Ethics Committee, South Australian Department for Health and Wellbeing, approval number HREC/18/SAH/42.

Thirty-two participants expressed an initial interest in becoming citizen scientists, with 20 signing up for a training workshop. Due to a variety of personal circumstances (illness, travelling and timing of the workshops), only 15 citizen scientists completed the training. Most (12) took part in a 90-min face to face workshop, with two receiving training over the phone and one face to face individually as they were unable to attend the workshop. All participants were provided with an easy to use, brief manual as a follow-up to training. Participants were also offered email and phone support from the research team to deal with any issues during the audit process as well as tailored individual support based on real-time validation of the frequency and usage of the audit tool.

Before citizen scientists were introduced to the audit tool, they were asked to complete an introductory survey, also hosted in the Survey123™ platform. This survey, which collected user demographic information, was opened for data collection at the commencement of each training session and was secured thereafter given that it contained personal information related to the citizen scientist. At completion of the introductory survey, citizen scientists were assigned an ID for use with data collection. This further anonymized the audit data collection process but allowed demographic data to be linked with audit data.

During induction, for those who selected the online audit tool, the link to the survey was placed on the citizen scientist’s mobile phone home screen for ease of access– giving the impression of an “app”. Those using the paper-based audit tool were given a booklet where each audit could be completed on one page to make field observations more convenient. They were also provided with a digital camera with which to visually record their location and reply-paid envelopes for returning audit sheets and the camera. Capturing data in situ gave citizen scientists the opportunity to provide “real-time” data which limited the introduction of recall bias to the study.

### 3.3. Deployment

Citizen scientists were given between six and ten weeks to conduct audits, using the tools provided, of any public green spaces they visited as part of their daily activities. We did not assign any specific spaces to go to; the idea was to let the citizen scientists conduct the audit without changing their normal routines. This ensured that audits mainly took place within the “life spaces” of citizen scientists, places and spaces that were part of their everyday lives and that were meaningful to them. There was no restriction as to how many audits to complete or where to complete them. Some citizen scientists took the audit tool with them when they travelled and even conducted audits outside South Australia. There were four email reminders sent during the data collection period to encourage citizen scientists to continue with their audits, wish them Happy Christmas and inform them of the closing date for audit uploads as the end of the data collection period approached.

### 3.4. Interviews and Co-Analysis

When the data collection period ended, each citizen scientist was invited to take part in a 1:1 interview with the research team during which several elements of the project were discussed. As well as exploring the data itself, the research team was interested in each citizen scientist’s views of the process regarding recruitment, use of the online or paper audit tool, survey content, and thoughts on how the data could or should be analysed. Along with autonomy and direction over data collection, it is this element of engagement which sets citizen science apart from usual data collection. One could argue that respondents are generally part of the research process since the submission of their data is the substance of this type of research; however, participation in decision-making around data collection interpretation and analysis is not a usual component of research.

At the interview, each participant was given a folder which contained three sections: the first was the aggregated, de-identified data from all participants; the second was their collated individual data; and the third was each individual audit they had carried out. During the interview, it was then easy to compare the individual’s collated data with the aggregated dataset from all participants, drawing conclusions about consistent trends and outliers in the data. Discussing individual audits provided an opportunity to gain a deeper understanding of the thought processes and differing interpretations of public green spaces made by the citizen scientists. As this was an iterative process, it allowed the research team to build a deeper understanding of the data set.

### 3.5. Analysis

Audit data consisted of three key elements: (1) spatial data; (2) preliminary demographic survey and Survey 123™ audit data and (3) recorded and transcribed interview data. Analysis of data consisted of three different approaches:
Spatial data, based on geocoded audit points (linked to the home address of each citizen scientist from the preliminary demographic survey) extracted from ArcGIS Online, was analysed using ESRI’s spatial analysis software (ArcMap 10.6.1). In particular, mapping of spatial data focused on creating spider maps for individual participants (where their home address formed the centroid point and the audit locations formed points linked to this centroid) as these were considered the most appropriate way to view the life spaces of individual citizen scientists. Spatial audit data were also viewed by demographic variables for potential themes; for example comparing audits for people who lived alone, by age or gender, and by audit variables, for example: measuring distance from home location to audit sites for public green spaces accessed by walking, or comparing location attributes where the length of visit was stated as less than 15 min.Quantitative data from both the preliminary demographic survey with each citizen scientists and the public green space audits were analysed using SPSS Statistics Version 26. Due to the small number of participants, this provided descriptive statistics only.Qualitative data in the form of the photographs and open-ended text comments from the public green space audits and the transcribed interviews with citizen scientists were analysed using an inductive thematic approach using NVivo 12. An inductive approach to thematic analysis allows research findings to emerge from the frequent, dominant or significant themes inherent in raw data. This was considered particularly important for use with open text comments on the audits and with exploring the photographs where citizen scientist responses were organic and not guided by interviewer questions or interests.

## 4. Results

### 4.1. Study Participants

Of the 15 Citizen Scientists taking part in the pilot study, 12 were female and three were male; they ranged in age from 60 to 84 years with four aged 70 + years and the remainder aged 60–69 years. Three lived alone and ten lived with a partner or spouse while one citizen scientist lived with relatives other than a partner or spouse. Thirteen of the citizen scientists were living in the Greater Adelaide metropolitan region at the time of project; with two living in rural towns outside the city region. All were retired at the time of the pilot study. Thirteen were still driving and 12 considered their self-rated health to be good or very good.

In total, 15 participants submitted 264 audits over a three-month period; this varied from 6 to 47 audits for individual citizen scientists, with an average of 17.6 audits. Some citizen scientists began data collection in October 2018 with a rolling recruitment and induction until mid-December 2018. Data collection halted on 31 January 2019. Follow-up interviews were conducted with 12 participants in February and early March 2019, with three citizen scientists unavailable for interview at this time due to travel commitments or illness.

### 4.2. Use of the Audit Tool

All citizen scientists were regular and confident users of smart phones prior to the pilot study. Two participants elected to use the paper-based version of the audit tool even though they owned smart phones, while the remaining 13 elected to use their smart phones to do the audit. Of the two citizen scientists who chose to use the paper-based version, one (female, aged 83) chose this method because she was a keen photographer and preferred to take photographs for the audits with her digital camera. The second (female, aged 62) was not confident using her older mobile phone and was concerned her limited data allowance would not cope with the audit tool requirements. Both of these citizen scientists posted the audit forms back to the research team to be entered into the online system. One uploaded her audit photographs into Google Drive™ while the other brought the camera in and researchers retrieved the photographs. At the post-audit interview both said they would have been happy to enter their own data into an online system via their home computer had this been an available option, emphasizing their comfort levels with technology.

The research team allowed two hours for training workshops and 90 min for 1:1 training either face to face or via the telephone. This included time to go through the consent process for research, provide some background information on citizen science methods and approaches, uploading the online audit tool to participants’ smart phones, completing the background survey, going through the audit questions and options for answers, and some time to practice using the audit tool. In fact, the workshops were completed in less than one hour, with less than 30 percent of this time needed to install and trial the online audit tool to a level where participants felt confident using it. The three participants unable to attend the workshop who were trained over the phone or face to face managed to work through the information and training in less than 45 min.

All participants were offered telephone and email backup assistance during the data collection process. Only two queries were received from participants during the data collection period, both related to online connectivity and data usage rather than issues with the online audit tool itself. In addition, at the post-audit interviews, three citizen scientists suggested they had been unsure they were “doing it right” as the audit tool did not indicate to the user that uploads had been successful after pressing the submit button. However, it should be noted that data collected in the audit process had a very high completion rate (there were no skipped questions in any audits) and there were also high rates of photograph uploads and free text comments. Of the 264 audits, over 99 percent had at least one photograph uploaded (with just over 40 percent having a second photo), 95 percent had correctly used the geocoded location finder, and over 73 percent (*n* = 195) included a short, open-ended comment.

Post-audit interviews were structured so that citizen scientists were first asked about the usability of the audit tool. Overall, citizen scientists liked the questions that had been included in the survey. The only question that was generally thought irrelevant asked “how are you feeling today?” The general sentiment here was that if they were not feeling okay they would not be out doing an audit. The most difficult aspect of using the audit tool appears to have been using the “target” GPS locator. While most audits were geocoded correctly, users were unsure at the time of the audit that they were correct, and this seemed to cause a slight anxiety for a few citizen scientists. However, most agreed that, overall, the technology had been simple to use, the audits were easy and quick to complete, and they felt confident that unassisted training via an online video or training package and/or with a training manual would have been possible. Over half said they had enjoyed the immediacy of the data collection process (audits done in situ and data uploaded straight away) but some would have preferred to complete the audits in situ and then upload data later using Wi-Fi.

In the post-audit interviews, citizen scientists were also asked to comment on the amount of direction they were given in terms of what to audit and where. Some felt that they had understood the brief very well and were confident that they had managed to capture the themes of the pilot study. Others felt that they would have liked more direction regarding what to audit. Suggestions for more guidance included having a “checklist” of potential audit sites for future projects, others felt being able to view de-identified data through access to the project’s ArcGIS website, or being able to connect with other citizen scientists in the pilot, would have encouraged them to consider other spaces they could audit.

### 4.3. Overview of Audit Data

While the analysis of the audit data was not the focus of the pilot study, it does provide some insight into the approaches to data collection taken by citizen scientists. Audits were carried out in a wide variety of places under the broad remit of public green spaces. Half of all audits were carried out within 1.6 km (straight line distance) of the citizen scientist’s home, with the rest being a mixture of regular activities or outings as part of everyday life (walks with friends, visiting relatives, being on holidays or socializing with others). As such, it is considered that all audits reflected the natural life spaces of citizen scientists. Only 11 audits (0.4%) were classed as “other” or not being “a green space” (described by the categories of “very green”, “somewhat green”, “mixed bare and green”, “mostly bare” on the Likert Scale response). These included spaces such as cafés, the theatre or library, and shopping precincts. Of the other 253 audits, 70 percent, were considered very green or somewhat green. These public green spaces varied from more formal parks, gardens and streetscapes in local neighbourhoods (as seen in Figure 2a) to more natural forest or bushland settings (see Figure 2b).

Several key design elements of public green spaces were identified in the analysis of both audit free text comments and photographs through inductive coding using NVivo12. Seating received the most references (97), followed by street trees (96), natural bushland (93), park trees (87) and water (in terms of creeks, lakes, rivers and the ocean) (51). Citizen scientists were asked several questions about the spaces they were auditing, including “Why have you come to this location?”; “How long do you usually spend here?” and “How did you get here?”. By far the common response to the first of these questions was “On my way to somewhere else” (*n* = 128), with the next three most common responses being “To relax” (*n* = 58), “To exercise” (*n* = 54) and “To meet others” (*n* = 52). In accordance with the responses to this first question, the most frequent response to “How long do you usually spend here” was “Less than 15 min” (*n* = 99) followed by “15 to 30 min” (*n* = 48). When looking at mode of transport in relation to both of these responses the most usual form of mobility by far was “walked” (*n* = 132). These responses, along with the fact that just over half of all audits were less than 1.6 km from home, suggests audit data relates to local neighbourhood engagement, which was reinforced in some of the open comments associated with the audits:
“*I frequently pass these places when out walking or on my way to the shops or library. A combination of council and resident plantings make the route very pleasant to use*”*[ID133]*
“*My streetscape. I walk down this street a number of times a day on the way to shops and or beach nearby*”*[ID186]*

In terms of creating a series of neighbourhood audits indicative of use of local public green spaces, open text comments appeared to reflect on the attributes of their local neighborhoods, the elements of good design that appealed to them and the impact the built environment had on their sense of wellbeing, as highlighted by these comments:
“*I go out of my way to ride down this road- trees, birds and plants along tram line give street a calming uplifting effect*”*[ID199]*
“*The building design enables conversation. Easy to hear!*”*[ID193]*
“*I choose this bus stop in preference to others purely because of the trees, grass, flowers and birds*”*[ID133]*

### 4.4. Participant Reflections on Senior Citizen Science

The post-audit interviews with participants also offered the opportunity to reflect on the citizen science process. The most common response to the question regarding the most enjoyable aspect of the experience was that our citizen scientists took greater notice of their physical surroundings as a result of the audit process; in particular, photographing these public green spaces seemed to be the catalyst for a closer examination at their everyday life spaces.
“*It really brought home to me how I choose my routes and my activities according to how much green space I can walk through or stop and have a rest in*”*[ID118]*
“*Yes, it did make me think a little bit more about the environment and how it can be made more conducive to people walking and doing recreational activities*” *[ID101]*
”*I think it was that thing of being more aware of your physical environment – like if I was with a friend and he or she would be saying ‘hurry up’ and I’d say ‘no look at this…...yes it heightens your awareness*”*[ID110]*

As part of this reflection, citizen scientists also offered suggestions on design of the larger project. These suggestions fell into three clear themes: (1) more engagement by citizen scientists in co-designing the audit tool and methodology; (2) providing a greater range of roles for citizen scientists beyond data collection, including elements of data cleaning, analysis, co-design workshops and activities that use the audit results (for example re-development of case study sites in the community or working with design students to create models of innovative public green spaces) and (3) creating a variety of ways to engage citizen scientists in the whole project– for example, through e-newsletters; an interactive web page that includes chat functions and de-identified aggregated data; short text messages updating citizen scientists on data collection progress (both individual and team progress), and opportunities to provide feedback on publications, reports and other forms of dissemination.

## 5. Discussion

A pilot study on the use of a citizen science approach to explore influences of neighbourhoods, particularly public green spaces, on daily lives of older people has been conducted. An on-line audit tool installed on the smart phones of older people, as citizen scientists, was developed by the research team and trialed by 15 trained citizen scientists. The citizen scientists were involved not only in data collection but also in data interpretation and preliminary analysis, as well as contributing to a review of the audit tool and overall study. Several themes on the use of citizen science for understanding the life spaces of older people have emerged.

### 5.1. Comfort Levels with New Technology

Older people, as citizen scientists, demonstrated high comfort levels with technology and therefore comfort with technology should not be underestimated among an older population when designing research projects. This small pilot study showed that participants were capable and eager to use technology to engage with science; although it is important to still have alternative means for all older people to fully participate in citizen science. While restricted to a small participant group, this confidence with digital technology was evident through several components of this study. Firstly, the high selection rate for using the online audit tool with a smart phone, where even participants who used a paper-based version of the audit tool had smart phones and all participants were comfortable communicating with the research team by email during the audit process. Secondly, the training process required much less time and interaction than anticipated, with a high rate of participants suggesting at the post-audit interview that they would have been comfortable carrying out training via a video and/or using a training manual. Thirdly, the small number of participants who requested assistance throughout the data collection period suggests participants were comfortable with the technology and audit process.

While this study is indicative of the increasing comfort levels older adults have with technologies, it also highlights the improved user friendliness of new technologies and tools such as the Survey123™ audit tool. High levels of compliance in audit submissions (no skipped questions) and high rates of uploading of photos and optional open-ended text comments suggest the audit tool platform is both user friendly and suitable for the general public. However, there were some issues with the audit tool that need to be addressed. For example, the need to manually geocode current location, either by pressing the target button to find current location, or by manually moving the map to show locate current location, can mean one of the most critical elements of the data collection tool (knowing where the audit has been conducted) may become unreliable. Secondly, a simple message to let users know that the audit data has been successful submitted would give users more certainty that their audit was complete. However, while there are some minor issues with the Survey123™ platform the advantages of in situ data collection, time and date stamped geocoded data, being able to include both photographic and free text data, and the fact that the software is available in over 30 languages makes this an ideal tool for use with older citizen scientists.

Of course, this pilot engaged a small, self-selected group of adults interested in participating in citizen science and may not represent the general older population. Further work needs to be done with wider groups of older adults, including those with reduced mobility, greater frailty and/or poorer health, and from different cultural backgrounds to test both the potential and reliability of the audit tool. Understanding the relationship between ageing and the built environment—particularly the potential value of age friendly environments—is critical for all older adults, not just those who are technology adept, fit and active. Further pilots are being planned using the audit tool with more frail older populations under different circumstances—for example in The Netherlands and Poland we are hoping to trial the audit tool in 2020 by pairing frail older adults with gerontology students in order to audit local neighborhoods for age friendliness.

### 5.2. The Audit Tool as a Medium to Reflect on Public Green Spaces

Citizen scientists felt they had thought more deeply and reflected on their own engagement with the built environments they lived and interacted in through the audit process. We feel the audit tool gave older people a medium to look critically at their neighbourhoods and lived environments in order to reflect on and understand more fully what components of their neighbourhoods they liked and did not like, and not only what they utilised but also where and why. While the audit tool alone allowed for the collation of data on this usage, along with pictures and qualitative expressions of their neighbourhood public green spaces, it was the follow-up interview that offered the opportunity to reflect on this more fully, by looking at the neighbourhood audits as a whole, highlighting both spatial and behavioural patterns of usage.

Of note for this pilot, these patterns of interaction with public green spaces in local neighbourhoods may be seen as transition points or “green corridors”—a conduit to everyday life rather than necessarily destinations in themselves. This was aptly summed up by one citizen scientist: “*I’m passing in transit. I have been thinking that this is my contact with many green spaces rather than visiting. I will record more of these*” [ID102]. This suggests how we could think about the design of neighbourhood public green corridors—with better pathways, more opportunities to sit and rest, points of interest along the way (through art, play, exercise equipment, or mediums for interaction with others). This may create spaces that act as links for community points of interest and activity (shops, public amenities, libraries and other public facilities, transport links and so forth); encouraging older people to use public green spaces for social and civic engagement, incidental exercise and as a way of engaging with nature without necessarily being destinations in themselves.

### 5.3. The Value of Citizen Science

This pilot highlights that even a small number of citizen scientists can tell us a lot about the built environment they interact in because of the high volume of audits they can produce in a short amount of time. With very few prompts, 15 citizen scientists produced 264 neighbourhood audits in this pilot study, with most saying they felt they could have completed more in the same time frame with more reminders. This is far beyond the scope of a small university-based research team alone and highlights the effectiveness of citizen science for data collection. The added value of using an online tool is that data collection could be carried out anywhere and is not limited to the geographical location of the university-based research team. Citizen scientists could theoretically be based anywhere in the world and submit geocoded, date and time stamped audits online in their own language, enabling data to be collected from a wide variety of locations simply and effectively.

The over-riding reflection on citizen science for participants in this pilot study was that it was an engaging use of their time where they felt they were contributing something of value to science as well as potentially improving outcomes for the neighbourhoods in which they lived. They appreciated the opportunity to examine their own data, the de-identified aggregated dataset and to reflect on the research tools and processes through the post-audit interviews. Participants showed a keenness to be further engaged with future citizen science projects beyond just data collection, indicating that whenever possible they would like to be involved in all stages of future research projects. Citizen science projects should make the most of this enthusiasm and engage citizen scientists early in the process. As with all research teams, citizen scientists bring different skill sets and interests to projects and these assets should be utilised to the project’s advantage by offering a range of activities to be engaged in, including data cleaning and analysis; co-design and planning of improved neighbourhood spaces, and presentation and dissemination of research outcomes.

## 6. Conclusions

Citizen science is a valuable tool for the social sciences, in particular for exploring the built environment and life spaces of older people. While the tools used in this pilot study have not yet been tested for reliability or validity, the outcomes of the pilot study show that further testing and retesting of the audit tools is a worthwhile future exercise. Further work also needs to be done of trialing the audit tool with different population groups, including frailer and less mobile older people and older people from different cultural backgrounds and in different geographical locations.

While this pilot study has focused on public green spaces it is felt that a citizen science approach using an audit tool that focuses more broadly on age friendly neighbourhoods would provide an opportunity to evaluate age friendly communities from the perspective of older people. The number of age friendly neighbourhoods and communities have expanded rapidly worldwide in the past 10 years; yet little work has been done in this time evaluating the differences age friendliness makes to the lives of those living in these communities. Such an audit tool offers opportunities to collect and collate data from the unique perspective of older people in these communities. Most importantly, the insider knowledge of older people about their own neighbourhoods has shown to be a valuable contribution to social science through the conceptual learning and deeper observation that citizen science offers. As cities and neighbourhoods around the world continue to adopt age friendly principles for the built environment, citizen science projects such as this pilot study offer sound approaches to evaluating and understanding the value of these approaches for creating better places to age well.

## Figures and Tables

**Figure 1 healthcare-07-00126-f001:**
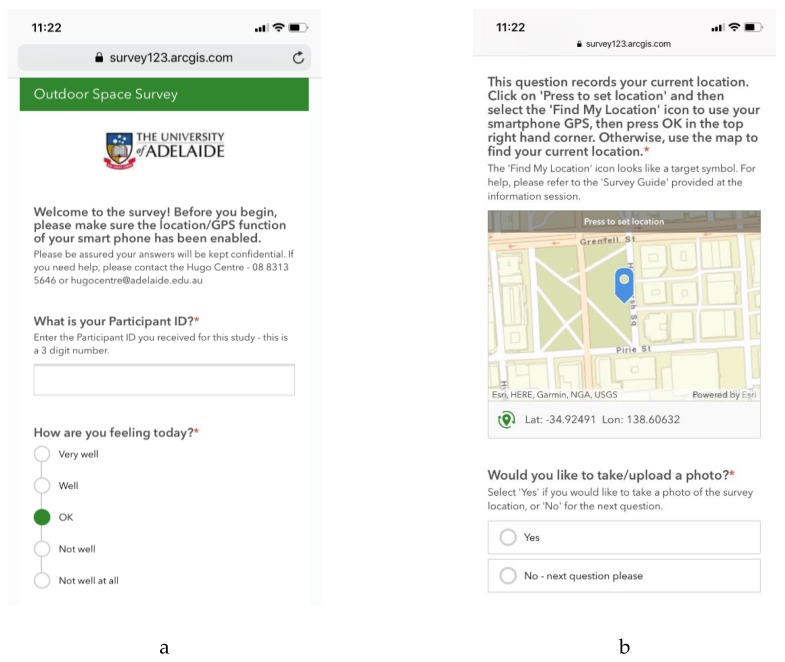
Outdoor Space Audit Tool (screen capture from a smart phone): (**a**) shows the opening screen of the audit tool; (**b**) shows the location data and photograph questions; (**c**) and (**d**) show examples of the questions from the audit tool.

**Figure 2 healthcare-07-00126-f002:**
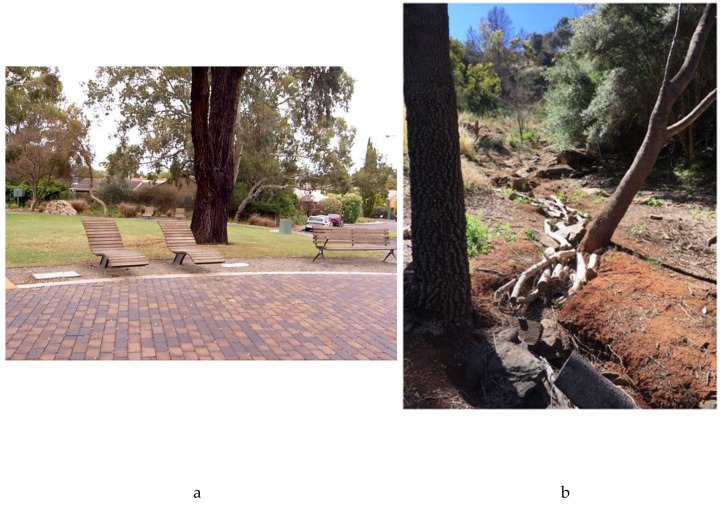
Examples of public green space images uploaded with audits. Figure (**a**) shows a more formal park space while (**b**) highlights some of the more natural “bushland” spaces.

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
