# Peer review of "Using Citizen Science to Explore Neighbourhood Influences on Ageing Well: Pilot Project"

_healthcare, 2019, doi:10.3390/healthcare7040126_

Round 1
Reviewer 1 Report
Introduction/abstract
The authors are using the term intimate knowledge, I would rather use 'insider knowledge'.
How do you define open and public spaces? Definition is needed in the Background or Methods sections. And please be consistent with using the terms ''open, indoor, green, public''. Throughout the article i have noticed that they were mixed up.
Background
Please try to make the link between age-friendly cities part, expert-determined approach and citizen science
Methods
3.2. Recruitment and training
How many of the participants have chosen online tool and how many of them paper-based?
Results and Discussion
It is stated that "Follow-up interviews were conducted with 12 participants 202 in February and early March 2019". What happened with the other three participants?
And please move the Demographics section to the Materials and Methods. It makes much more sense to read it before the Results and Discussion. Also the other sections of the Results (eg Demographics, Use of the audit tool) should be implemented in to the Materials and methods.
Please restructure the Results and Discussion. There is no Discussion now and the Results are scattered and not well integrated. There are no main themes and the codes showed from NVivo are not needed, you should integrate all the themes emerged from qualitative and quantitative parts together.
Moreover there is also no explanation about how you have performed the analysis (eg how qualitative and quantitative data was analyzed). What approach was used for analyzing qualitative data? Grounded theory? Something else?
The Conclusions also have to be integrated into the Discussion and not listed with bullet points. Then you can make some general conclusions in the Conclusions section if needed.
Author Response
Thank you for these valuable and clear suggestions which has helped to create a stronger paper. Changes are as follows:
Change ‘intimate knowledge’ to ‘insider knowledge’.
Three changes were made to provide consistency throughout the paper.
Insert a definition of open and public spaces into the introduction or background section.
A definition has been inserted into the introduction section (lines 41-47)
Be consistent with the use of the terms open, green and public spaces
The term ‘public green spaces’ has been used consistently throughout the paper to avoid confusion.
Make a stronger link between age friendly cities, expert determined approaches and citizen science in the background.
Additional text has been added to the background section linking age friendly cities and citizen science (lines 80-84)
In the methods section state the number of participants who chose the online audit tool versus the paper version.
This information can be found in the results section (lines 247-250)
In the results and discussion section explain why three participants did not take part in a follow-up interview.
This information has been included in the results section (lines 238-240)
Move the demographics section to the ‘Materials and Methods’ section.
This has been done (see lines 230-236)
Restructure the results and discussion section to create separate results and discussion sections – with discussion organised by themes.
The results now form section 4 of the paper, with a separate discussion (section 5). The discussion section utilises the dot points from the previous conclusion section as themes for the discussion (see suggestion point 10 below) with a separate conclusion
Explain analysis approaches.
The analysis section has been expanded to explain the various components more fully (see section 3.5)
Integrate the conclusion bullet points into the discussion section and then add in a general conclusion.
See explanation at point 8 above, a short concluding paragraph has now been developed as section 6 of the paper.
Reviewer 2 Report
The paper reports on the use of citizen science to explore neighbourhood influences on ageing well. The authors describe a pilot project in which they trialed the citizen science approach using a co-created model in which older people not only collected data but were also engaged in preliminary analysis of the data and contributed feedback and ideas on the methods, process, audit tool used and the design of the project. This is an innovative approach that allows the appraisal and perception of residents themselves regarding the quality and daily experience of open space (something that is lacking in the standardized ‘expert assessments’).
There is no question regarding the relevance and importance of these kind of innovative research practices. However, due to the ‘newness’ of the methodology, some questions remain unanswered.
Page 3 + 10:
The authors are describing that they developed two audit tools. However, some crucial information regarding the strategy of the development process is missing. I will try to illustrate my thought process while reading the manuscript:
How are these audit tools exactly developed? Was there a guideline followed when developing the tools? Where the domains covered in the tools based on a literature review? Expert opinion? Other instruments?
Furthermore, there is a lot of focus on the usability and feasibility of the audit tools. However, are reliability and validity considered during the development process? Especially considering the strong statements in the conclusions ‘citizen science has proven to be a valuable method for exploring the built environment and life spaces of older people’. Can the authors really say this based on this pilot study including 15 citizen scientists? And are the authors sure that the audit tools totally cover the concept of the ‘built environment’ (content + construct validity)?
I understand that the authors cannot cover everything at once. However, some further clarification or at least consideration is needed here in my opinion.
Page 6:
The analysis paragraph could be improved, as it is unclear what type of analysis is performed. What is exactly done with the spatial data? The quantitative data is analyzed using SPSS. However, are you only doing descriptive analyses, or performing any tests? It becomes clear later, yet some clarification is in order here. Furthermore, the coding process for the photographs and qualitative data should be elaborated on (who did the coding, how, etc.). Perhaps the authors could check the COnsolidated criteria for REporting Qualitative research checklist).
Page 10 (Conclusions)
As mentioned before, the general conclusion (line 371) seems somewhat too ‘positive’. Although these kind of innovative research approaches should be initiated more, we should be careful not to accept them without rigorous scientific evaluation (in a larger research project).
In conclusion, I really enjoyed reading the manuscript, and believe that after some clarification and adjustments it is perfectly suitable for publication in Healthcare.
Author Response
Thank you to reviewer two for the insightful and helpful directions for improving the paper. The following changes have been made:
Insert into the methods section the strategy in developing the audit tools – i.e.: did we follow any guidelines, use other domain tools, develop the tools based on literature or other instruments?
An explanation of the strategy used has been inserted into section 3.1 of the methods section (lines 116/117)
Discuss the reliability/validity of these audit tools during development.
References has been made to this in both the Methods section (3.1, lines 117/119) and in the conclusion (lines 454/456)
Remove the phrase ‘citizen science has proven…’ from the conclusion.
This has been done
The paper needs more realistic conclusions taking into consideration that it is a small pilot study
This has been done – references to the limitations to the study have been added in the discussion section and in the conclusion to the paper
Round 2
Reviewer 1 Report
The topic of the article is interesting and relevant. The authors endeavored all the suggested changes.